# (HTBNet)Arbitrary Shape Scene Text Detection with Binarization of Hyperbolic Tangent and Cross-Entropy

**DOI:** 10.3390/e26070560

**Published:** 2024-06-29

**Authors:** Zhao Chen

**Affiliations:** School of Remote Sensing and Information Engineering, Wuhan University, Wuhan 430072, China; oliver@whu.edu.cn

**Keywords:** scene text detection, binarization, hyperbolic tangent, MSCA, FMCS, cross-entropy

## Abstract

The existing segmentation-based scene text detection methods mostly need complicated post-processing, and the post-processing operation is separated from the training process, which greatly reduces the detection performance. The previous method, DBNet, successfully simplified post-processing and integrated post-processing into a segmentation network. However, the training process of the model took a long time for 1200 epochs and the sensitivity to texts of various scales was lacking, leading to some text instances being missed. Considering the above two problems, we design the text detection Network with Binarization of Hyperbolic Tangent (HTBNet). First of all, we propose the Binarization of Hyperbolic Tangent (HTB), optimized along with which the segmentation network can expedite the initial convergent speed by reducing the number of epochs from 1200 to 600. Because features of different channels in the same scale feature map focus on the information of different regions in the image, to better represent the important features of all objects in the image, we devise the Multi-Scale Channel Attention (MSCA). Meanwhile, considering that multi-scale objects in the image cannot be simultaneously detected, we propose a novel module named Fused Module with Channel and Spatial (FMCS), which can fuse the multi-scale feature maps from channel and spatial dimensions. Finally, we adopt cross-entropy as the loss function, which measures the difference between predicted values and ground truths. The experimental results show that HTBNet, compared with lightweight models, has achieved competitive performance and speed on Total-Text (F-measure:86.0%, FPS:30) and MSRA-TD500 (F-measure:87.5%, FPS:30).

## 1. Introduction

Krizhevsky [1] and LeCun [2] designed convolutional neural networks, which laid the foundation for modern deep learning. There have been great achievements, especially in computer vision, such as 3D vision [3,4] and automatic vision [5]. Recently, deep-learning-based [6,7,8] methods of scene text detection [9,10] have made rapid progress due to their wide range of practical applications such as license plate recognition [11], automatic driving [12], smart city [13] and password guessing [14]. Scene text detection [15,16] can be regarded as a special kind of object detection task [17,18,19] due to the following aspects: (1) Compared with general object detection, text in scenes has the characteristics of arbitrary shape, which is hard to be detected by a rectangular box. (2) In the meantime, scene text is always with various fonts and colors, so it is more complex than object detection. Therefore, scene text detection has always been a challenging task.

Generally speaking, scene text detection is mainly divided into three methods based on regression, component, and segmentation. Among them, the regression-based scene text detection methods greatly draw on the method of general object detection. By presetting or generating a large number of proposal boxes of different sizes and shapes at different positions in the image, the regression-based models obtain the bounding boxes containing text instances through training parameters and finally adjust the shape and position of the bounding boxes to obtain the final text area. The regression-based scene text detection method network model has no complex post-processing reasoning speed, but its detection accuracy is generally inferior to the segmentation method.

Component-based scene text detection methods disassemble text into individual components and then use relational inference to assemble the scattered components into characters. Component-based methods work well for long text, despite a relatively high loss in disassembly and reassembly.

Segmentation-based scene text detection methods are pixel-level classification and post-processing, which can more accurately describe scene text of arbitrary shape and irregular arrangement. However, most of the existing segmentation-based scene text detection methods [20,21,22,23,24,25] require complex post-processing. On the one hand, the inference speed is slow, making it difficult to be used in industrial applications. On the other hand, the post-processing operation is independent of the training process, which greatly reduces the detection performance of the network. The differentiable binary network model [26] in the early stage successfully solved this problem. The post-processing operation was integrated into the training process, and the overall training and optimization were carried out, which greatly improved the detection performance of the network and achieved SOTA at that time. However, the convergence speed of the model is slow. At the same time, the sensitivity to features in different channels and scales is lacking, so it is easy to miss texts of different scales.

Aiming at the above-mentioned two problems, we contrive the text detection Network with Binarization of Hyperbolic Tangent (HTBNet). Specifically, we design the Binarization of Hyperbolic Tangent (HTB), which greatly improves the convergence speed. At the same time, the Multi-Scale Channel Attention (MSCA) and Fused Module with Channel and Spatial (FMCS) are proposed. For feature maps in different scales, the features of the two dimensions in channel and spatial are integrated to further improve the detection performance. Due to different channels of feature maps at the same scale focusing on different object regions, the MSCA module assigns different weights to different channels and adds to the network training process to adjust and optimize. Objects of different scales in an image are difficult to simultaneously and sufficiently detect, and so FMCS combines features of different scales, which improves the sensitivity of feature scales, and can obtain both backbone and detail features. Combining the above two modules makes the detection performance better.

In summary, our main contributions are three-fold:We propose the Binarization of Hyperbolic Tangent (HTB), leading the convergence speed during training from 1200 epochs to 600 epochs.We design the cross-entropy loss function, which is differentiable, enabling the use of optimization algorithms such as gradient descent to minimize the loss function.We contrive the Multi-Scale Channel Attention (MSCA) and the Fused Module with Channel and Spatial (FMCS), which interfold features from different scales in channel and spatial. Our method achieves outstanding results on Total-Text and MSRA-TD500 benchmarks.

## 2. Related Work

With the rapid development of deep learning [2,27,28,29,30,31,32], scene text detection has also made great progress both in the academic and industrial fields. Generally speaking, deep-learning-based text detection methods can be subdivided into three categories: regression-based methods, component-based methods, and segmentation-based methods.

### 2.1. Regression-Based Methods

Regression-based [33,34,35,36,37,38] methods usually enjoy simple post-processing algorithms (e.g., non-maximum suppression [39]). However, most of them are limited to representing accurate bounding boxes for irregular shapes, such as curved shapes. For example, EAST [38] consisted of two stages. In stage one, a fully convolutional network extracted text regions. In stage two, non-maximum suppression [39] (NMS) was used to remove unsuitable text predictions. It could detect text at any orientation and its speed was very fast. But its accuracy was not particularly high. To improve accuracy, there was MSR [37], which was an evolved version of EAST. It introduced a multi-scale network and a novel shape regression technique for predicting dense text boundary points. These boundary points enabled the precise localization of scene text with different orientations, shapes, and lengths. There are another series of methods that are based on improvements to SSD [40], such as TextBoxes [34], TextBoxes++ [33], and DMPN [35]. TextBoxes [34] drew inspiration from the SSD [40] (Single Shot MultiBox Detector) for text detection. It modifies the SSD architecture by replacing fully connected layers with convolutional layers and adapting the convolutional kernel size 3 × 3 to 1 × 5 to handle text detection better, considering the different aspect ratios of text compared to general objects. Additionally, TextBoxes used a different set of default box aspect ratios compared to SSDs, typically incorporating ratios like 1, 2, 3, 5, 7, and 10 to account for the wide variety of text aspect ratios. It employed a single deep neural network for both text detection and character-level regression, enabling efficient and fast text detection. TextBoxes [34] could only detect horizontal text, while TextBoxes++ [33] extended this capability to detect multi-oriented text. This improvement involved the following three key changes: (1) Aspect Ratios of Default Boxes: the aspect ratios of the default boxes were modified to include 1, 2, 3, 5, 1/2, 1/3, and 1/5, enabling the detection of text with different aspect ratios. (2) Convolutional Kernel Size: the 1 × 5 convolutional kernel was changed to 3 × 5 for generating text box layers, which helped improve text detection. (3) Multi-Oriented Text Box Output: TextBoxes++ was designed to output multi-oriented text boxes, allowing it to detect text at various angles. Another method based on SSDs was DMPN [35], which was designed for horizontal text detection. DMPN [35] achieved multi-oriented text detection by learning the position information of four points relative to multi-oriented anchors. ATTR [36] was based on the Faster R-CNN [31], a classical regression-based text detection method. ATTR [36] was a two-stage text detection method. The first stage was similar to Faster R-CNN, utilizing CNN, RPN, and ROI to obtain text proposals. The second stage involved refining these text proposals to make the predicted boxes more accurate. Regression-based methods aim to fit text boundaries, and overall, they are not as accurate as segmentation-based methods.

### 2.2. Component-Based Methods

The component-based [41,42,43,44,45] scene text detection method involved text regions into individual components, using relationship inference to identify components belonging to the same text line. Then, appropriate post-processing techniques to obtain the text regions were used. DRRG [45] employed a graph neural network to model the geometry and relationships of text, aiming for high-precision text detection. This method was capable of accommodating various text shapes, including horizontal, vertical, and curved text, making it highly applicable in the field of scene text detection. CRAFT [41] used a segmentation method that differed from traditional image segmentation. Unlike pixel-level classification for the entire image, CRAFT only predicted character centers. It consisted of two branches. One was focused on the probability of character centers, and the other was focused on character-to-character connection relationships. After post-processing, the text bounding boxes were obtained. There is also a series of methods, which are based on SSDs [40], such as SegLink [42] and SegLink++ [43]. The core of SegLink [42] was to transform text detection into the detection of two local elements: segment and link. The segment was a directional box that covered a part of the text content, while the link connected two adjacent segments, expressing whether these two segments belonged to the same text. The algorithm merged relevant segments into the final bounding box based on the representation of links, improving detection accuracy and training efficiency. SegLink++ [43] built upon the original SegLink [42] by introducing two types of lines: attractive links and repulsive links. These two types of lines connected segments belonging to the same text region and kept segments from different text regions apart, respectively, achieving better detection results. Component-based methods work well for long texts, but the limitation lies in the loss associated with splitting and recombining.

### 2.3. Segmentation-Based Methods

The segmentation-based [20,21,22,23,24,25,26,46,47,48,49] scene text detection method performs pixel-wise binary classification for text and background, followed by complex post-processing to obtain the final text regions. SAE [22] embedded shape awareness and separated closely placed text instances. It addressed the problem of excessively long text lines by clustering the output of the three results. PAN [24] achieved arbitrary-shaped scene text detection through segmentation principles and offered both speed and scale invariance. PixelLink [20] introduced two pixel-wise predictions based on DNN: text/non-text prediction and link prediction. TextSnake [21] predicted the Text Center Line (TCL) and text regions (TRs), acquiring a general representation of the scene text of arbitrary shapes. DBNet [26], based on segmentation, obtained the threshold map and the probability map. It proposed differentiable binarization, simplifying post-processing and achieving high-precision real-time text detection. DBNet++ [46], based on DBNet [26], added the Adaptive Scale Fusion module, leading to higher precision. However, the DBNet [26] had a slow convergence speed during training for 1200 epochs, and there was a possibility for improvement in feature extraction from the backbone and neck. Therefore, we propose HTBNet to achieve faster convergence during training. Additionally, we design MSCA and FMCS to thoroughly integrate the features from different scales in channel and spatial, thereby improving the accuracy of the model.

## 3. The Proposed Method

### 3.1. Overview

The overall structure of HTBNet designed in this paper is illustrated in Figure 1, which consists of four components (Backbone, MSCA, FMCS, and HTB). The backbone extracts features from the input image, and then the neck (MSCA, FMCS) further processes and fuses these features. Finally, the detection heads (HTB) predict text regions based on the fused features, and post-processing is applied to obtain the final text regions. Four components of the proposed HTBNet are the following: (1) ResNet50 [28] is adopted as the backbone to gain multi-scale feature maps. (2) MSCA fuses the multi-scale feature maps obtained from the backbone. (3) FMCS simultaneously integrates information of fused feature maps from MSCA in channel and spatial. (4) HTB inputs the feature maps obtained by FMCS into the corresponding prediction head to obtain the probability map and threshold map. Then, our designed hyperbolic tangent function computes the values in the probability map and threshold map to obtain the initial binary map. The final text regions are obtained after post-processing of the binary map. The structure details of the MSCA, FMCS, and HTB modules are explained in Section 3.2, Section 3.3, and Section 3.4, respectively.

### 3.2. Multi-Scale Channel Attention (MSCA)

The initial input of HTBNet is a scene text image, and ResNet50 serves as the backbone network to extract features, resulting in five feature maps at different scales, corresponding to 1/2, 1/4, 1/8, 1/16, and 1/32 of the original image size. Due to feature maps from different channels at the same scale focusing on different regions, we propose the novel MSCA module to better represent the kernel features of all objects in the image. The MSCA module is essentially a kind of channel attention mechanism that adds trainable weights across feature maps from different channels at the same scale. It has improved model expressiveness in the early computation of the three classical computer vision tasks (image classification, object detection, and semantic segmentation). As shown in Figure 1, the input to MSCA is five feature maps at different scales, with sizes being 1/2, 1/4, 1/8, 1/16, and 1/32 of the original image. Now, the same operations are applied to each of these five feature maps. First, the feature maps at the same scale undergo global average pooling, resulting in a tensor with a dimension equal to the number of the feature map’s channels. Then, it passes through two fully connected layers, squeezing the dimension of this tensor to one-sixteenth, and then expanding it back to its original dimension. Finally, the new tensor multiplies channel-wise with the original feature maps. In Figure 1, ‘up×N’ represents upsampling the feature map N times, and ‘down×1/2’ represents downsampling the feature map to 1/2. Additionally, ‘concat’ represents concatenation operation. The above computation process can be more intuitively expressed using Equations (1)–(3) as follows:(1)$$C_{i}=F_{ave}(M_{i}),\;\left(i=1,\;2,\;3,\;4,\;5\right)$$
where $F_{ave}$ is the global average pooling in the spatial dimension, and $M_{i}(i=1,\;2,\;3,\;4,\;5)$ represents the feature maps at one specific scale as the input of the MSCA module; meanwhile, $C_{i}(i=1,\;2,\;3,\;4,\;5)$ represents a tensor with a dimension equal to the amount of corresponding $M_{i}$ channels as the output of Equation (1).
(2)$$D_{i}=F_{linear}(C_{i}),\;\left(i=1,\;2,\;3,\;4,\;5\right)$$
Here, $F_{linear}$ performs two consecutive fully connected layers, and $D_{i}(i=1,\;2,\;3,\;4,\;5)$ is a tensor that has the same size as $C_{i}(i=1,\;2,\;3,\;4,\;5)$.
(3)$$N_{i}=F_{product}(M_{i},D_{i}),\;\left(i=1,\;2,\;3,\;4,\;5\right)$$
Here, $F_{product}$ refers to the element-wise product, equal to $M_{i}$ multiplying the $D_{i}’s$ corresponding value of each channel by all pixel values in the spatial dimension of the corresponding channel, and $N_{i}(i=1,\;2,\;3,\;4,\;5)$ has the same size as $M_{i}(i=1,\;2,\;3,\;4,\;5)$. Then, the multi-scale feature maps, $M_{i}(i=1,\;2,\;3,\;4,\;5)$ are operated with concatenation, resulting in the new feature maps, Q, whose size is 1/4 of the original image’s size.

According to the above operations of the MSCA module, subsequent ablation experiments have shown that, with little increase in the number of model parameters, significant improvement in model performance has been achieved.

### 3.3. Fused Module with Channel and Spatial (FMCS)

MSCA aggregates features from different channels at the same scale, but to simultaneously detect objects at different scales, we have designed FMCS to aggregate features at different scales and positions. The upper branch of FMCS merges channel information, leading to aggregating the features from all channels at different scales in the whole image. The lower branch of FMCS fuses spatial information by taking the average over the channel dimension of the feature maps, gaining a feature map with one channel. Then the feature map undergoes three convolutional operations, while kernel size is 3, and padding is 1. Next, the feature map is expanded channel- and element-wise and multiplied with the feature map obtained from the upper branch. Finally, the new feature map is added element-wise to the feature map obtained from the upper branch, resulting in the final feature map of FMCS. It is important to emphasize that, element-wise product and element-wise addition, respectively, represent element-wise multiplication and element-wise addition for two feature maps totally at the same shapes. The above process can be expressed as Equations (4) and (5) as follows:(4)$$E=F_{c\_ave}(Q)$$
where $F_{c\_ave}$ refers to the global average pooling in the channel dimension, and Q represents the feature maps as both the output of the MSCA module and the input of the FMCS module; meanwhile, as the output of Equation (4), E is a feature map of the same size as Q in spatial dimension and E has only a single channel. The functions of $F_{ave}$, $F_{linear,}$ and element-wise product is the same as Equation (1), Equation (2), and Equation (3), respectively. Element-wise addition can be described as Equation (5).
(5)$$K=F_{add}(G,H)$$
Here, $F_{add}$ refers to element-wise addition, and specifically, it is that the pixel values at the corresponding positions of G and H are added element-wise to form a new feature map K. Subsequent ablation experiments indicate that the FMCS module significantly improves the model’s performance.

### 3.4. Binarization of Hyperbolic Tangent (HTB)

Based on the new feature K, obtained from the previous feature fusion, we design the HTB module to enable end-to-end training and fast convergence of the model. The probability map (P_map_ ) and threshold map (T_map_) are obtained from the corresponding prediction heads. HTB module performs exponential operations based on the difference between the values corresponding to the probability map and the threshold map using the hyperbolic tangent function. P_map_ and T_map_ undergo computation using our designed hyperbolic tangent function (Tanh) to obtain the initial binary map (B_map_). As is shown in Equations (6) and (7) as follows, the hyperbolic tangent function (Tanh) is an integral part of the HTB module:(6)$$binary\_base=\frac{e^{m}-e^{-m}}{e^{m}+e^{-m}}$$
(7)$$m=k\left(P-T\right)$$
where P and T represent the values of the corresponding pixels on the probability map and threshold map at the same position, respectively. k is the super-parameter, and we set it to 50. The binary_base is the initial value of the feather map. If binary_base is greater than 0, then the corresponding pixel is considered to belong to the text region; otherwise, it belongs to the background region. Tanh’s diagram compared with the sigmoid function, used in the baseline, is shown in Figure 2a.

As seen in the subsequent experimental section, by using the hyperbolic tangent function (Tanh), the training process that initially converged in 1200 epochs is shortened to 600 epochs, significantly improving the model’s convergence speed. From a mathematical perspective, we can analyze the reason for the faster convergence of the model. As is well known, deep learning models involve the process of backpropagation, where first-order partial derivatives are calculated for various weight coefficients. These first-order partial derivatives are then multiplied by the learning rate to obtain the corresponding weight coefficient decrement, as shown in Equation (8).
(8)$$w^{\prime }=w-lr*\frac{\partial L}{\partial w}$$
Here, L represents the total loss function, while lr represents the learning rate. And w is the initial weight coefficient, and $w^{\prime }$ is the updated weight coefficient corresponding to it. Based on the theoretical foundation above, we calculate the first-order derivatives of the two functions, and the resulting diagram is shown in Figure 2b. It is evident that Tanh’s grad is steeper than the sigmoid’s grad, which means that for the same variable step, the function value of the hyperbolic tangent function changes more significantly than the sigmoid. Subsequent experiments prove that the weight coefficients of the hyperbolic tangent function decay faster than those of the sigmoid, resulting in faster convergence of the overall model.

Finally, post-processing is applied to the initial text regions obtained by HTB, involving expansion and contraction, to obtain the final text regions. It is worth noting that during the inference phase, initial text regions can be obtained using only the probability maps or threshold maps, without the necessity to compute the hyperbolic tangent function.

### 3.5. Cross-Entropy Loss Function

Based on the earlier computation of the probability map, the binary map, and the threshold map, by comparing the predicted values with the ground truth, we can obtain the corresponding loss functions, respectively. The predictions of the probability map and the threshold map in this article are typical classification problems. Cross-entropy [50] loss function can effectively guide the model to adjust parameters, making the predicted values gradually approach the ground truths. Therefore, we use the cross-entropy loss function for both probability map loss ($L_{s}$) and binary map loss ($L_{b}$). In subsequent experiments, the loss functions of mean square error and cross-entropy have been compared, confirming that the cross-entropy loss function exhibits superior performance.

The loss function L can be expressed as a weighted sum of the probability map loss ($L_{s}$), the binary map loss ($L_{b}$), and the threshold map loss ($L_{t}$), which refers to the specific Equations (9)–(11).
(9)$$L=L_{s}+L_{b}+10\times L_{t}$$
(10)$$L_{s}=L_{b}=\sum_{i\in M}y_{i}logx_{i}+\left(1-y_{i}\right)log\left(1-x_{i}\right)$$
Here, M represents a set where positive samples and negative samples are in the ratio of 1:3 and $x_{i}$,$y_{i}$ represent the ground truth and prediction value of the probability map or binary map, respectively.
(11)$$L_{t}=\sum_{i\in N}\left|y_{i}^{*}-x_{i}^{*}\right|$$
Here, N represents a set of pixels within the text bounding boxes and ${x_{i}^{*},y}_{i}^{*}$ represent the ground truth and prediction value of the threshold map, respectively. With the loss functions, the entire network can undergo backpropagation and gradient computation, enabling the optimization of parameters.

## 4. Experiments and Results Analysis

### 4.1. Datasets and Evaluation

This paper focuses on natural scene text images of arbitrary shapes, so we utilize the Total-Text [51] and MSRA-TD500 [52] datasets for experimental purposes. Examples of the two datasets are shown in Figure 3 and Figure 4. Additionally, before formal training, we conducted pre-training on the SynthText [53] synthetic dataset, which is shown in Figure 5. The datasets involved in this paper are described below.

Total-Text (Curved Text Dataset): This dataset primarily consists of English text with a smaller portion of Chinese text. It includes 1255 images in the training set and 300 images in the test set. The text in this dataset is often curved, and it is annotated at the word level using polygons.

MSRA-TD500 (Multi-Oriented Scene Text Dataset): This dataset is focused on multi-oriented text detection and includes both Chinese and English text. It comprises 300 images in the training set and 200 images in the test set, with text annotated at the text-line level.

SynthText, which consists of 800k images, is a synthetic dataset used for training and evaluating text detection and recognition models. It comprises computer-generated text placed on a variety of backgrounds to simulate real-world text scenarios. This dataset offers a wide range of text appearances, including different fonts, sizes, orientations, and background textures. We utilize SynthText to pre-train our model, enhancing its ability to detect text in diverse, real-world environments.

Scene text detection is a crucial task in the field of computer vision, aiming to accurately identify and locate text regions within images captured in natural scenes. To assess the performance of text detection algorithms, quantitative analysis is often conducted using metrics such as precision, recall, and F1 Score. Precision is the proportion of correctly identified positive samples out of all samples predicted as positive by the model, which is calculated using Equation (12) as follows:(12)$$Precision=\frac{TP}{TP+FP}$$
where TP (True Positives) represents the number of samples correctly identified as positive, and FP (False Positives) represents the number of samples incorrectly identified as positive. Precision measures the accuracy of the model’s positive predictions, with higher values indicating better precision in positive predictions.

Recall is the proportion of correctly identified positive samples out of all actual positive samples, which is calculated using Equation (13) as follows:(13)$$Recall=\frac{TP}{TP+FN}$$
where TP represents the number of samples correctly identified as positive, and FN (False Negatives) represents the number of actual positive samples incorrectly identified as negative. Recall measures the model’s ability to identify actual positive samples, with higher values indicating broader coverage of actual positives.

The F1 Score is the harmonic mean of precision and recall, providing a balanced assessment of a model’s precision and recall. It is calculated using Equation (14). The F1 Score ranges between zero and one, with higher values indicating a better balance between precision and recall. These three evaluation metrics play a crucial role in natural scene text detection. Precision focuses on the accuracy of positive predictions, recall assesses the coverage of actual positive samples, and the F1 Score combines both aspects to offer a comprehensive evaluation of model performance. In practical applications, the choice of evaluation metrics depends on task requirements, and sometimes a balance between precision and recall needs to be considered. The comprehensive consideration based on precision and recall leads to the use of the common F1 Score as the primary evaluation metric in this paper. On the other hand, there is another metric to measure the computational efficiency of the model, which is Frames Per Second (FPS). FPS measures the speed of the algorithm in processing natural scene images, indicating the number of frames that the deep learning algorithm can handle per second. A high FPS indicates that the algorithm has high real-time performance, making it suitable for practical application scenarios.
(14)$$F1\;Score=\frac{2Precision*Recall}{Precision+Recall}$$

The experiments in this paper use the Ubuntu 20.04 operating system and PyTorch 1.12.0 deep learning framework. The training platform utilizes an NVIDIA GeForce RTX 3090 Ti graphics card with 24 GB of VRAM. We first pre-train the model with the SynthText dataset for 100k iterations. Then, we finetune the models on the corresponding real-world datasets for 600 epochs. It is known that there are 1200 epochs in the original baseline. The decay strategy of the learning rate that we adopt in this paper is SGD. The learning rate is set as Equation (15) as follows:(15)$$lr=lr*{\left(1-\frac{iter}{max\_iter}\right)}^{0.9}$$
where lr is the learning rate, the initial value of which is set to 0.007, and iter represents the current iteration times; meanwhile, max⁡_iter represents the maximum iteration times.

### 4.2. Ablation Study

We conduct a series of ablation experiments on two datasets (i.e., Total-Text and MSRA-TD500) for each of the three proposed modules. The results of the ablation experiments for datasets Total-Text and MSRA-TD500 are presented in Table 1, Table 2, Table 3 and Table 4.

On the one hand, HTB accelerates the convergence speed of the model by speeding up the gradient descent. On the other hand, as can be seen from Table 1, Table 2, Table 3 and Table 4, HTB significantly enhances performance. HTB has resulted in an improvement of 0.7% and 0.2% of F-measure for the Total-Text dataset when using Res18 and Res50 as the backbone, respectively. And the HTB module increases by 0.3% and 0.7% of F-measure for the MSRA-TD500 dataset. What is more important is that HTB reduces the training process from the initial 1200 epochs to 600 epochs, greatly shortening the training time.

The MSCA module, which is used to fuse the features in channel dimension, leads to more accurate features. MSCA has exhibited a significant improvement on the two datasets and the two backbone networks, with F-measure increasing by a minimum of 0.5% and a maximum of 0.9%.

FMCS aims to fuse features in both spatial and channel dimensions simultaneously. According to above Table 1, Table 2, Table 3 and Table 4, FMCS achieves the highest improvement in F-measure. FMCS results in an improvement of 0.7% and 1.3% of F-measure for the Total-Text dataset while Res18 and Res50 are adopted as the backbone. And FMCS increases by 2.5% of F-measure for the MSRA-TD500 dataset whether the backbone is Res18 or Res50.

The model, incorporating all three modules, can obtain the overall results. While we use Res18 as the backbone, the F-measures improve by 1.3% on both Total-Text and MSRA-TD500. Likewise, while we use Res50 as the backbone, there is a greater improvement of 2.6% on both Total-Text and MSRA-TD500.

To further validate the performance of the cross-entropy loss function, we compared it with the mean square error loss function. The specific experimental results are shown in Table 5. It can be seen that the convergence speed of the training process for the cross-entropy loss function has been reduced from 1200 epochs to 600 epochs compared to the mean square error loss function. Meanwhile, the values of F have improved. I trained the model with one 3090Ti GPU card, which took about 50 h of training for 600 epochs.

To validate the values of *K* = 50 in Equation (7), and a multiplication factor ×10 in Equation (9), we set them as different values. As shown in Table 6 and Table 7, we can find that *K* = 50 in Equation (7) and the multiplication factor ×10 in Equation (9) achieve the best performances both with the backbone of ResNet18 or ResNet50.

By randomly selecting several images from the test set for single-image testing, we can obtain visual results as shown in Figure 6 below. Especially from the two images on the left and in the middle, it can be seen that our method significantly outperforms the baseline. However, for the image on the right, due to the difficulty in distinguishing between the font and the background, both methods exhibit relatively poor detection performance.

### 4.3. Comparisons with Other Advanced Methods

We also compare our method with the previous advanced methods, and the results are presented in Table 8 and Table 9. It can be observed that when we utilize Res50 [28] as the backbone network, HTBNet achieves F-measures of 86% and 87.5% on datasets Total-Text and MSRA-TD500, respectively, outperforming the performance of the previously mentioned methods in Table 8 and Table 9. At the same time, to make the model more lightweight, we also use Res18 [28] for comparison. The model speed was significantly improved, with FPS increasing from the initial 30 to 49 and 56 on Total-Text and MSRA-TD500, respectively, and for the detection performance, the F1 Score remains competitive. According to Table 9, we plot a two-dimensional scatter plot of performance versus speed on MSRA-TD500, as shown in Figure 7. The horizontal axis represents the FPS of the model, and the vertical axis represents the F1 Score of the model. From both the performance and speed perspectives, our model achieved the best results.

In the field of text detection, looking back at lots of previous work, there has been a usual focus on the model’s performance, which is equal to the F1 Score, while the computational complexity of the model is frequently overlooked. However, the lightweight of deep learning models holds significant value and profound significance in practical applications in the industry. Firstly, lightweight can reduce the computational and storage resource requirements of the model, enabling more efficient deployment and operation in resource-constrained environments such as embedded devices and mobile devices. Secondly, lightweight contributes to improving the model’s inference speed, reducing response time, thereby enhancing real-time capabilities, which is suitable for applications requiring rapid response, such as the scene text detection task in this paper. Additionally, lightweight can lower the energy consumption of the model, extending the device’s battery life, which is crucial for battery-powered applications like mobile devices and drones. Overall, research and application of lightweight in deep learning models can propel the penetration of artificial intelligence technology into a broader range of fields, realizing more intelligent, efficient, and sustainable applications.

As shown in Figure 8, HTBNet_res50 (ours) compared to DB_res50 (baseline) indicates that our model essentially converges at 600 epochs, while the DB_res50 converges at 1200 epochs. This is consistent with the inference conclusions from Figure 2.

As shown in Figure 9, the precision–recall curves of HTBNet_res50 (ours) and DB_res50 (baseline) indicate that the HTBNet_res50’s curve completely encloses the DB_res50’s curve, leading to the conclusion that our model performs better.

## 5. Conclusions

In this paper, we have contrived a novel framework for detecting arbitrary-shape scene text, which improves the performances of text detection from three aspects: (1) The HTB module is proposed to integrate the post-processing process into the training period and accelerate the model’s convergence during the training. (2) The designed cross-entropy loss function accurately describes the difference between the predicted values and the ground truths, which improves the model performance. (3) The proposed MSCA and FMCS extract and fuse features from channel and spatial dimensions, enhancing the model’s ability to perceive objects of different scales and positions. All of the three modules significantly improve text detection accuracy. The experiments have verified that our HTBNet consistently outperforms outstanding methods in terms of speed and accuracy.

## Figures and Tables

**Figure 1 entropy-26-00560-f001:**
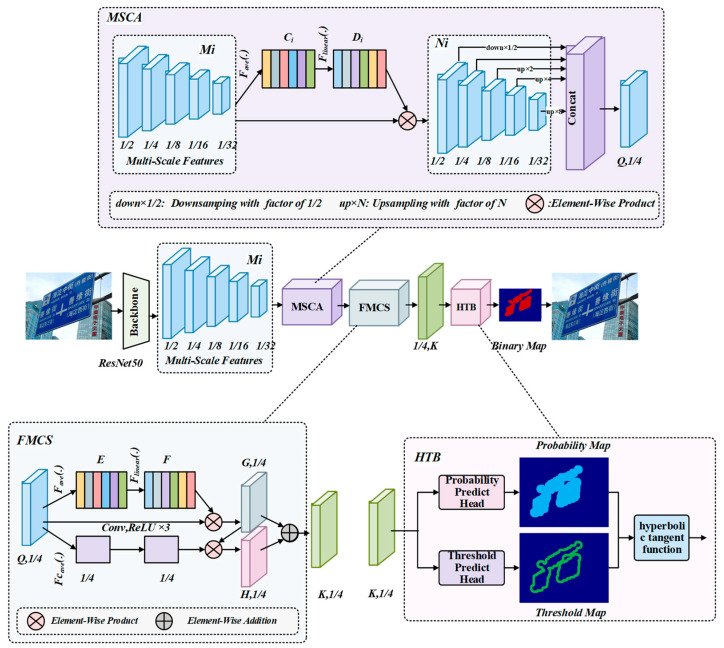
The architecture of the proposed HTBNet.

**Figure 2 entropy-26-00560-f002:**
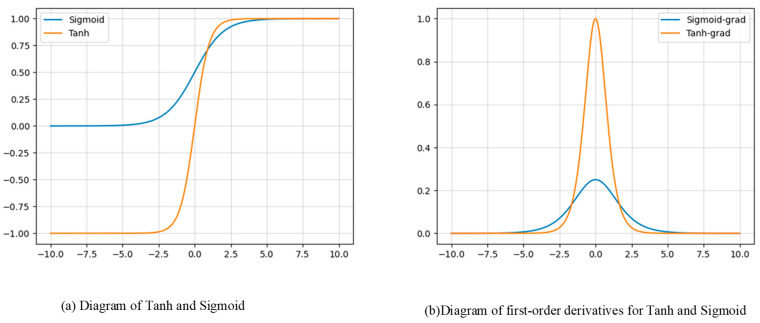
The diagrams of the two functions and corresponding first-order derivatives.

**Figure 3 entropy-26-00560-f003:**
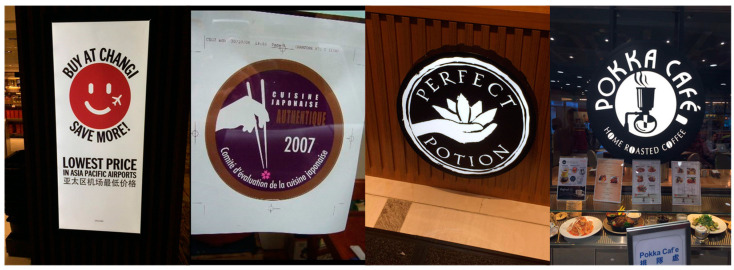
Examples of Total-Text.

**Figure 4 entropy-26-00560-f004:**
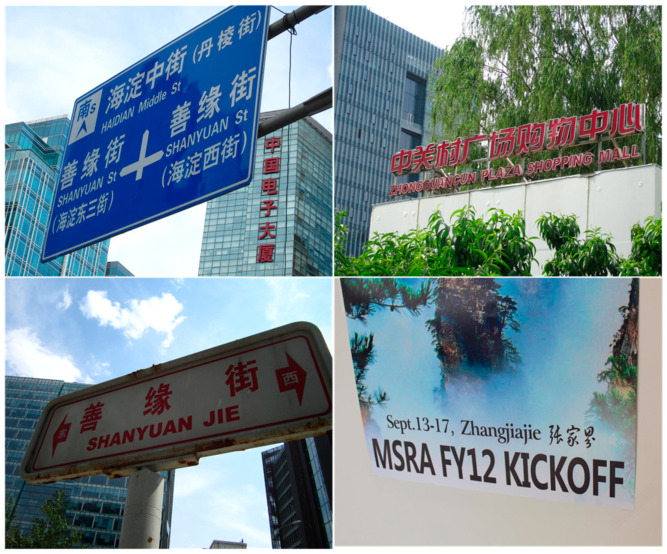
Examples of MSRA-TD500.

**Figure 5 entropy-26-00560-f005:**
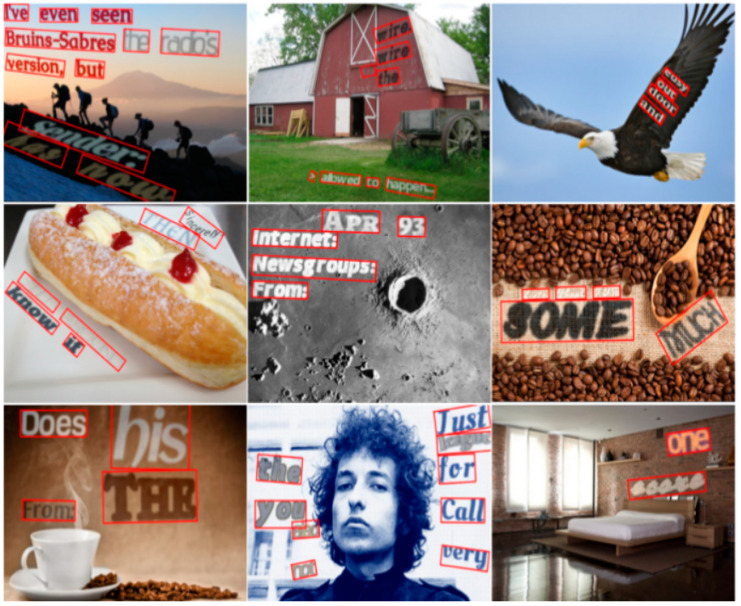
Examples of SynthText [53].

**Figure 6 entropy-26-00560-f006:**
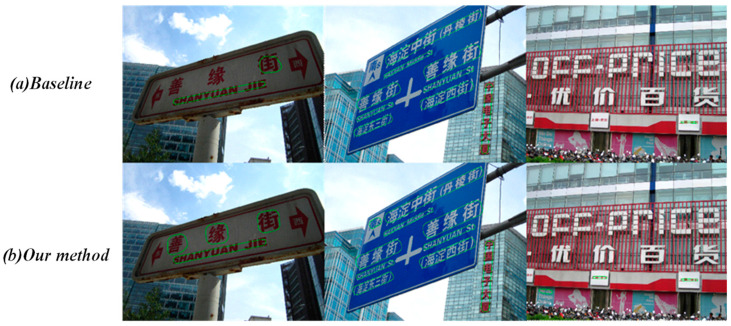
Detection results of (**a**) baseline and (**b**) our method.

**Figure 7 entropy-26-00560-f007:**
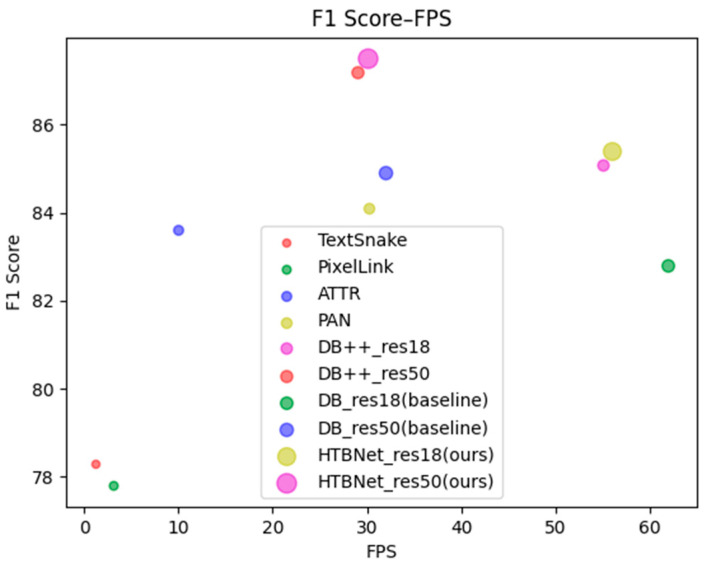
The two-dimensional scatter plot of performance versus speed on MSRA-TD500.

**Figure 8 entropy-26-00560-f008:**
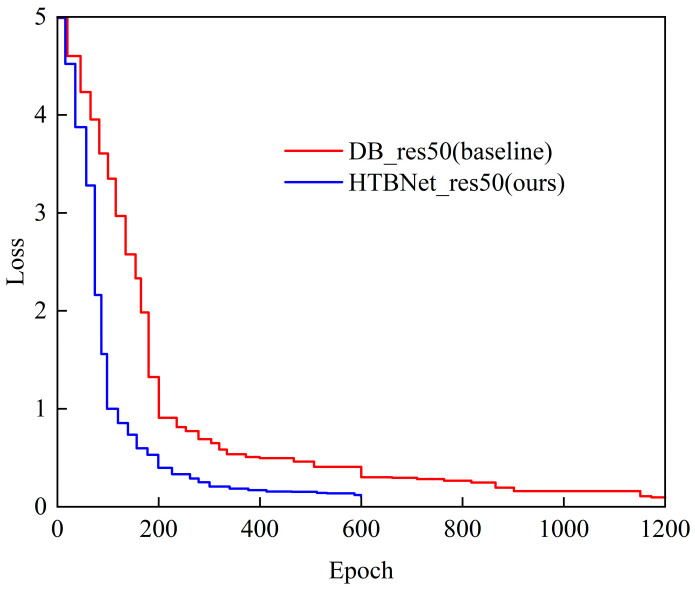
The loss–epoch curves on Total-Text.

**Figure 9 entropy-26-00560-f009:**
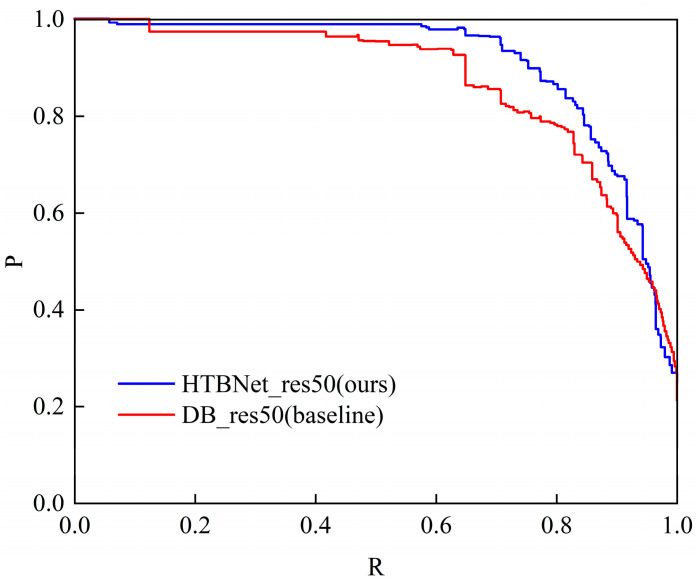
The precision–recall curves on Total-Text.

**Table 1 entropy-26-00560-t001:** Detection ablation results on Total-Text with backbone of ResNet18. ‘P’, ‘R’, and ‘F’ represent, separately, precision, recall, and F-measure.

Module	HTB	MSCA	FMCS	Total-Text		
				P	R	F
DB_res18 (baseline)	✕	✕	✕	88.3	77.9	82.8
res18	✓	✕	✕	**90.9**	77.1	83.5
res18	✕	✓	✕	89	78.9	83.7
res18	✕	✕	✓	88.5	79	83.5
HTBNet_res18 (ours)	✓	✓	✓	86.8	**81.6**	**84.1**

**Table 2 entropy-26-00560-t002:** Detection ablation results on Total-Text with backbone of ResNet50.

Module	HTB	MSCA	FMCS	Total-Text		
				P	R	F
DB_res50 (baseline)	✕	✕	✕	87.1	82.5	84.7
res50	✓	✕	✕	**94.9**	76.8	84.9
res50	✕	✓	✕	87.9	82.8	85.3
res50	✕	✕	✓	90.5	**81.3**	**86**
HTBNet_res50 (ours)	✓	✓	✓	91.3	**81.3**	**86**

**Table 3 entropy-26-00560-t003:** Detection ablation results on MSRA-TD500 dataset with backbone of ResNet18.

Module	HTB	MSCA	FMCS	MSRA-TD500		
				P	R	F
DB_res18 (baseline)	✕	✕	✕	90.4	76.3	82.8
res18	✓	✕	✕	89.3	77.7	83.1
res18	✕	✓	✕	**92.3**	75.9	83.3
res18	✕	✕	✓	88.8	**82**	85.3
HTBNet_res18 (ours)	✓	✓	✓	89.8	81.4	**85.4**

**Table 4 entropy-26-00560-t004:** Detection ablation results on MSRA-TD500 dataset with backbone of ResNet50.

Module	HTB	MSCA	FMCS	MSRA-TD500		
				P	R	F
DB_res50 (baseline)	✕	✕	✕	91.5	79.2	84.9
res50	✓	✕	✕	90.3	81.4	85.6
res50	✕	✓	✕	89.7	82.3	85.8
res50	✕	✕	✓	91.9	**83.3**	87.4
HTBNet_res50 (ours)	✓	✓	✓	**92.2**	**83.3**	**87.5**

**Table 5 entropy-26-00560-t005:** Detection ablation results in loss functions.

Module	Mean Square Error	Cross-Entropy	MSRA-TD500				Total-Text			
			P	R	F	Convergence Epochs	P	R	F	Convergence Epochs
HTBNet_res18	✓	✕	87.9	82.5	85.1	1200	87.5	80.7	84	1200
HTBNet_res18 (Ours)	✕	✓	89.8	81.4	85.4	600	86.8	81.6	84.1	600
HTBNet_res50	✓	✕	91.5	83.5	87.3	1200	90.5	81.5	85.8	1200
HTBNet_res50 (Ours)	✕	✓	92.2	83.3	**87.5**	**600**	91.3	81.3	**86**	**600**

**Table 6 entropy-26-00560-t006:** Detection ablation results on hyper-parameters with backbone of ResNet18.

Module	K	Multiplication Factor	MSRA-TD500			Total-Text		
			P	R	F	P	R	F
HTBNet_res18 (Ours)	10	3	89.3	81.2	85.1	86.6	81.1	83.8
HTBNet_res18 (Ours)		10	89.5	81.3	85.2	86.8	81.4	84.0
HTBNet_res18 (Ours)		30	89.6	81.0	85.1	87.0	79.7	83.2
HTBNet_res18 (Ours)	50	3	89.5	81.2	85.1	86.6	81.5	84.0
HTBNet_res18 (Ours)		10	89.8	81.4	**85.4**	86.8	81.6	**84.1**
HTBNet_res18 (Ours)		30	90.1	80.9	85.3	87.0	81.1	83.9
HTBNet_res18 (Ours)	250	3	89.5	81.0	85.0	86.4	81.6	83.9
HTBNet_res18 (Ours)		10	89.7	81.2	85.2	86.6	81.6	84.0
HTBNet_res18 (Ours)		30	90.1	80.5	85.0	86.9	81.0	83.8

**Table 7 entropy-26-00560-t007:** Detection ablation results on hyper-parameters with backbone of ResNet50.

Module	K	Multiplication Factor	MSRA-TD500			Total-Text		
			P	R	F	P	R	F
HTBNet_res50 (Ours)	10	3	91.8	83.1	87.2	91.0	80.8	85.6
HTBNet_res50 (Ours)		10	92.0	83.3	87.4	91.2	81.0	85.8
HTBNet_res50 (Ours)		30	92.2	82.8	87.2	91.5	80.5	85.6
HTBNet_res50 (Ours)	50	3	92.0	83.2	87.4	91.1	81.0	85.8
HTBNet_res50 (Ours)		10	92.2	83.3	**87.5**	91.3	81.3	**86.0**
HTBNet_res50 (Ours)		30	92.5	82.8	87.4	91.6	80.7	85.8
HTBNet_res50 (Ours)	250	3	92.0	82.5	87.0	90.5	81.2	85.6
HTBNet_res50 (Ours)		10	92.1	83.1	87.4	90.8	81.4	85.8
HTBNet_res50 (Ours)		30	92.3	82.6	87.2	91.2	80.6	85.6

**Table 8 entropy-26-00560-t008:** Detection results on Total-Text dataset. “P”, “R”, and “F” indicate precision, recall, and F1 Score, respectively. * represents the result is unknown.

Methods	P (%)	R (%)	F (%)	Params (M)	FPS
TextSnake [21]	82.7	74.5	78.4	218.9	*
PixelLink [20]	53.5	52.7	53.1	218	*
ATTR [36]	76.2	80.9	78.5	*	*
SAE [22]	82.7	77.8	80.1	*	*
PAN [24]	89.3	81	85	46.8	39.6
MSR [37]	73	85.2	78.6	*	*
DRRG [45]	84.9	**86.5**	85.7	198.6	*
DenseTextPVT [49]	89.4	80.1	84.7	*	*
DB++_res18 [46]	87.4	79.6	83.3	55.9	48
DB++_res50 [46]	88.9	83.2	86	116.3	28
DB_res18 (baseline) [26]	88.3	77.9	82.8	55.3	**50**
DB_res50 (baseline) [26]	87.1	82.5	84.7	115.7	32
HTBNet_res18 (ours)	86.8	81.6	84.1	55.5	49
HTBNet_res50 (ours)	**91.3**	81.3	**86**	115.8	30

**Table 9 entropy-26-00560-t009:** Detection results on MSRA-TD500. * represents the result is unknown.

Methods	P (%)	R (%)	F (%)	Params (M)	FPS
TextSnake [21]	83.2	73.9	78.3	218.9	1.1
PixelLink [20]	83	73.2	77.8	218	3
ATTR [36]	82.1	85.2	83.6	*	10
SAE [22]	84.2	81.7	82.9	*	*
PAN [24]	84.4	83.8	84.1	46.8	30.2
MSR [37]	76.7	87.4	81.7	*	*
DRRG [45]	82.3	**88.1**	85.1	198.6	*
PCBSNet [47]	90	76.7	82.8	*	*
TDGCN [48]	89.7	85.1	87.4	*	*
DB++_res18 [46]	87.9	82.5	85.1	55.9	55
DB++_res50 [46]	91.5	83.3	87.2	116.3	29
DB_res18 (baseline) [26]	90.4	76.3	82.8	55.3	**62**
DB_res50 (baseline) [26]	91.5	79.2	84.9	115.7	32
HTBNet_res18 (ours)	89.8	81.4	85.4	55.5	56
HTBNet_res50 (ours)	**92.2**	83.3	**87.5**	115.8	30

## Data Availability

The Total-text and MSRA-TD500 datasets are available at https://opendatalab.com/OpenDataLab/TotalText, accessed on 27 October 2017 and https://opendatalab.com/OpenDataLab/MSRA-TD500, accessed on 26 October 2012, respectively.

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
