# Peer review of "(HTBNet)Arbitrary Shape Scene Text Detection with Binarization of Hyperbolic Tangent and Cross-Entropy"

_entropy, 2024, doi:10.3390/e26070560_

Round 1
Reviewer 1 Report
Comments and Suggestions for Authors
The article is well written, content is logical, the structural framework is complete. However, the following comments need to be addressed:
1. Bulk of papers are cited without mentioning the contribution of each in specified area. It is suggested to clearly mention the contribution of each in the specified area.
“Recently, deep-learning-based[1-7] methods of scene text detection[8-11] “
“Generally speaking, scene text detection is mainly divided into three methods based on regression[24-29], component[30-34], and segmentation[35-42].”
2. Authors used K=50 in Eq.7, a multiplication factor x10 in Eq. 9 but did not provided any evidence that why such constant values are required.
3. The contribution in terms of convergence time reduction is not clear. How the time is reduced from 1200 to 600 epochs. What are the main parameters that reduce the convergence time.
4. Authors specified that their main contribution is to reduce the convergence time. However, in result section no convergence curves and no detail results are provided for speeding the training, validation, and prediction.
5. Authors can refer to the below papers results section and provide the comparison:
a. Number of trainable parameters for proposed and the benchmarks
b. Training time, epochs, and convergence curves
[1]. A Vehicle-Edge-Cloud Framework for Computational Analysis of a Fine-Tuned Deep Learning Model
[2]. A Deep Learning-Based Framework for Offensive Text Detection in Unstructured Data for Heterogeneous Social Media
Reviewer 2 Report
Comments and Suggestions for Authors
In this paper, dealing with scene text detection, authors developed the Binarization of Hyperbolic Tangent (HTB), in order to enhance the convergence speed.
Additionally, they introduced the Multi-Scale Channel Attention (MSCA) and Fused Module with Channel and Spatial (FMCS).
These modules integrate features across different scales in both channel and spatial dimensions to boost detection performance.
The MSCA module assigns varying weights to different channels, optimizing network training. The FMCS combines multi-scale features to enhance sensitivity and capture both backbone and detailed features.
The results are convincing and clearly presented.
Authors should address the following suggestions:
- Which pre-processing denoising methods are required for the proposed technique?
- Can you clarify why text detection is a special kind of object detection task?
- Authors should elaborate in more details and corroborate the claim in the Conclusion: ''The designed cross-entropy loss function accurately describes the difference between the predicted values and the ground truths, which improves the model performance.''
- The paper :
Nicoletta Saulig, Miloš Milovanović, Siniša Miličić, Jonatan Lerga, Signal useful information recovery by overlapping supports of time-frequency representations, IEEE transactions on signal processing 70, 5504-5517, 2022.
also deals with segmentation, thresholded coefficients and complexity maps. Can the two techniques be somehow linked?
Comments on the Quality of English LanguageThe English language is adequate.
Round 2
Reviewer 1 Report
Comments and Suggestions for Authors
The authors have addressed all the concerns and accommodated all the comments.